# Identification of Clinical Isolates of *Candida albicans* with Increased Fitness in Colonization of the Murine Gut

**DOI:** 10.3390/jof7090695

**Published:** 2021-08-27

**Authors:** Rebeca Alonso-Monge, Daniel Prieto, Ioana Coman, Sara Rochas, David M. Arana, Susana Hidalgo-Vico, Elvira Román, Jesús Pla

**Affiliations:** 1Departamento de Microbiología y Parasitología-IRYCIS, Facultad de Farmacia, Universidad Complutense de Madrid, Avda. Ramón y Cajal s/n, 28040 Madrid, Spain; adprieto@ucm.es (D.P.); icoman@ucm.es (I.C.); sararochaslopez@gmail.com (S.R.); shvico@ucm.es (S.H.-V.); elvirarg@ucm.es (E.R.); 2Servicio de Microbiología, Hospital Universitario de Getafe, 28905 Getafe, Spain; dmolinaa@salud.madrid.org

**Keywords:** commensalism, adaptation, mycobiota, *Candida albicans*, murine gut, fitness

## Abstract

The commensal and opportunistic pathogen *Candida albicans* is an important cause of fungal diseases in humans, with the gastrointestinal tract being an important reservoir for its infections. The study of the mechanisms promoting the *C. albicans* commensal state has attracted considerable attention over the last few years, and several studies have focused on the identification of the intestinal human mycobiota and the characterization of *Candida* genes involved in its establishment as a commensal. In this work, we have barcoded 114 clinical *C. albicans* isolates to identify strains with an enhanced fitness in a murine gastrointestinal commensalism model. The 114 barcoded clinical isolates were pooled in four groups of 28 to 30 strains that were inoculated by gavage in mice previously treated with antibacterial therapy. Eight strains that either exhibited higher colonization load and/or remained in the gut after antibiotic removal were selected. The phenotypic analysis of these strains compared to an RFP-tagged SC5314 wild type strain did not reveal any specific trait associated with its increased colonization; all strains were able to filament and six of the eight strains displayed invasive growth on Spider medium. Analysis of one of these strains, CaORAL3, revealed that although mice required previous bacterial microbiota reduction with antibiotics to be able to be colonized, removal of this procedure could take place the same day (or even before) *Candida* inoculation. This strain was able to colonize the intestine of mice already colonized with *Candida* without antibiotic treatment in co-housing experiments. CaORAL3 was also able to be established as a commensal in mice previously colonized by another (CaHG43) or the same (CaORAL3) *C. albicans* strain. Therefore, we have identified *C. albicans* isolates that display higher colonization load than the standard strain SC5314 which will surely facilitate the analysis of the factors that regulate fungal colonization.

## 1. Introduction

*Candida albicans* is the most prevalent fungal pathogen in humans world-wide. It can cause either superficial or systemic infections and among these, invasive candidiasis are an important health problem and are frequently linked to invasive procedure in medicine [1]. *C. albicans* colonizes the skin and mucosal surfaces of around 70% of the human population [2,3] and there is no significant life of this fungus outside the mammalian host; therefore, being colonized by *C. albicans* is a prerequisite for a subsequent infection [4]. Colonization takes place during infancy [5] and the same strain remains as a commensal for years as revealed by longitudinal molecular typing studies [6]. The main cause of candidiasis are commensal strains being the gut an important source of infections [7,8]. Prolonged antibiotic treatment, alterations of the immune system or disruption of the natural barriers result in translocation through intestinal barriers to the blood stream [9,10].

The use of murine commensalism models has allowed for the identification of some genes relevant for the establishment of the fungus as a commensal. As laboratory mice are not naturally colonized by *C. albicans,* different approaches have been used to allow *C. albicans* colonization. These make use of gnotobiotic mice [11,12], immunocompromised adult mice [10] or animals with a reduced microbiota due to the use of a wide spectrum antibiotic therapy [13,14,15,16]. Many different genes have been shown to mediate gut colonization [17,18,19]. These genes encode transcription factors [15], MAP kinases [14] or enzymes involved in metabolic routes [20,21] or morphogenetic transitions [11,22,23,24,25]. Although many of these studies have made use of deletion mutants, gene overexpression strains have also been studied. Thus, the study of a conditional overexpressing gene collection allowed for the identification of the transcription factor Crz2 as an important regulator of early adaptation to the murine gut [26] or even the regulator of the white-opaque switch, Wor1 [22,27]. Some phenotypic traits have been related to colonization, such as bile salts tolerance, adhesion to different surfaces, oxygen-dependent metabolism and cellular morphology.

We have previously reported a murine model with partially depleted bacterial microbiota and the use of fluorescence tagged *C. albicans* strains to perform competitive fitness experiments in the same animal by both culture-dependent and independent detection methods [14]. Culture-dependent methods allowed us to discriminate between two strains due to the red color developed by the RFP (Red Fluorescent Protein) carrying strain on Synthetic Define solid medium. Pool analyses of several strains are also possible: Perez and co-workers have made use of barcoded transcription factor mutant library to perform analysis in mouse models of intestinal colonization and disseminated infection [15]. Similarly, Znaidi and co-workers barcoded a collection of tetracycline-inducible gene overexpression strains [26]. These strategies require the extraction of DNA of fungal cells and perform qPCR with barcode specific primers to estimate the abundance of each mutant. NGS (Next Generation Sequencing) methodologies are powerful tools to analyze microbiota but also to identify pathogens in patients at risk [28] or genetic markers of antifungal resistance [29]. Here, we report the barcoding of a collection of *C. albicans* clinical isolates obtained from hospitalized patients or healthy donors to identify strains with enhanced ability to colonize the murine gastrointestinal tract using NGS. In total, 8 out of 114 *C. albicans* strains established as a commensal and remained with high fungal loads despite removal of antibiotic therapy. One of these strains was further characterized being a promising strain for improving gut commensalism model in mice.

## 2. Materials and Methods

### 2.1. Strains and Growth Conditions

The clinical *C. albicans* isolates were supplied by the Getafe University Hospital or obtained from healthy volunteers at the Department of Microbiology and Parasitology of the Faculty of Pharmacy of the Universidad Complutense de Madrid (UCM). These strains were barcoded with a DNA fragment that differs in 6 pb and that allows their further identification (Appendix A). Other *C. albicans* used in this work are listed in Table 1:

Yeast strains were routinely grown in YPD liquid medium (2% glucose, 2% peptone, 1% yeast extract) at 37 °C in an orbital shaker. Growth was estimated by OD_600_ measurements. YPD plates supplemented with chloramphenicol to 20 µg/mL was used for counting fungal CFUs. To count CFUs in RFP-labelled and unlabeled *C. albicans* strain mixtures, SD-chloramphenicol (2% glucose, 0.5% ammonium sulphate, 0.17% yeast nitrogen base supplemented with amino acids, 2% agar and 20 µg/mL chloramphenicol) plates were used.

### 2.2. Genetic Procedures

To barcode the *C. albicans* clinical isolates, we generated a collection of plasmids derived from pDUP3 vector [30]. This plasmid carries the *SAT1* dominant marker and a 550 bp flanking region of homology to the *NEUT5L* intergenic region that facilitates integration in this region of the *C. albicans* genome that does not affect cell physiology. The following degenerated primers: o-barcode-up (GGCCGCGTAGATCTGACCGTCGNNNNNNTACGTACTGCACGTAT) and o-barcode-lw (CGATACGTGCAGTACGTANNNNNNCGACGGTCAGATCTACGC) were used to generate a doble stranded DNA fragment leaving *Not* I and *Cla* I cohesive ends (N represent random degeneracy at that position). Both primers were mixed at the same concentration and exposed to 90 °C for 10 min followed by 5 min at 60 °C and 5 min at 50 °C to allow homologue pairing. The double chain DNA obtained with cohesive ends was introduced in the *Not* I-*Cla* I pDUP3 digested plasmid. We incorporated a *Bgl* II internal restriction site in the sequence (underlined) to identify those plasmids that integrated the barcode. Individual pDUP3-barcode plasmids were sequenced to generate and ordered the barcoded vector collection. One hundred fourteen *C. albicans* strains (as well as SC5314 strain) were transformed with individual plasmids after *Sfi* I digestion to direct the integration at the *NEUT5L locus* by electroporation [31]) and transformants were selected on YPD solid plates supplemented with 200 µg/mL of nourseothricin. The correct integration was confirmed by PCR using the primers: o-seqpDUP3up (GCTTGATATCCCGCGGTGGAGC) and o-complw (CACGAATCGTTAATAAGCTGTGATTGC) and sequencing.

The pNIM1R-dTOM2 [14] plasmid was used to generate fluorescent SC5314-RFP and CaORAL3-RFP labelled strains. This plasmid allows a tight tetracycline regulation (repressible) of the RFP fluorescent gene (TET-OFF system) and carries the *SAT1* dominant marker. The plasmid pNIM1R-dTOM2 was digested with *Kpn* I and *Ksp* I restriction enzymes and integrated at the *ADH1* region of *C. albicans* strains. *C. albicans* strains were similarly transformed by electroporation and transformants were selected on YPD supplemented with 200 µg/mL of nourseothricin.

### 2.3. In Vivo Procedures

The protocol used in the colonization model was approved by the Animal Experimentation Committee of the UCM (CEA 33-2015) and Comunidad de Madrid according to the Artículo 34 del RD53/2013 (PROEX 226/15). This model does not result in disease and all procedures were conducted to minimize mice suffering. Mice euthanasia was performed by CO_2_ inhalation following standard protocols (AVMA Guidelines for the Euthanasia of Animals: 2013 Edition). The number of animals per experiment was adjusted to the minimum for ethical reasons. Female mice C57BL/6 obtained from Charles River RMS, Spain, were housed in sterile cages with unlimited access to sterile food and water, and used within an age of 7–10 weeks-old. Mice housing and other non-invasive procedures took place in the animal facility from the Medical School of the UCM.

The gut colonization assay was performed following the protocol described previously [32] with minor modifications. Briefly, after 4–10 days of antibiotic pre-treatment (2 mg/mL streptomycin, 1 mg/mL bacitracin, 0.1 mg/mL gentamycin and 0.25 mg/mL fluconazole, fluconazole was removed one day before *C. albicans* inoculation), 10^7^
*C. albicans* cells were intra-gastrically introduced in a single inoculation by gavage. Antibiotic treatment was removed in most experiments when stated. In some experiments, a second inoculation was performed when stated. In the co-housing experiments, mice from different groups were put together in the same cage without performing any artificial inoculation. Colonization was determined by counting *C. albicans* CFUs in fresh stools obtained from each individual mouse at different time points. A 40 mg/mL feces suspension in sterile PBS was mechanically homogenized, serially diluted and cultured on YPD or SD-agar plates with chloramphenicol (20 μg/mL). Colonies were associated to a specific population in accordance to the colony color when RFP expressing strains where involved. To analyze *C. albicans* loads in the gastrointestinal tract, mice were sacrificed and samples from the stomach, cecum, small and large intestine were aseptically obtained, homogenized in sterile PBS and cultured in SD plates for CFUs counting.

### 2.4. NGS Assay

To perform the NGS assay, colonies from the stools of mice colonized with the pools of barcoded strains were collected and the genomic DNA was extracted. The amount of DNA was quantified in a nanodrop and equalized to generate the libraries for Illumina sequencing. A fragment of 272 pb flanking the barcode sequence was used as template for NGS. DNA libraries were prepared following minor modifications of the “16S Metagenomic Sequencing Library Preparation” [33] with adapted primers including locus specific sequences Calbtag Fw.2 (5′ CTTGATATCCCGCGGTGGAGC 3′) and Calbtag Rv (5′ GTGAGGGTTAATTTCGAGCTTGGCG 3′). Two rounds of amplifications were performed to add the Illumina adapters and library index to the DNA fragments. Illumina sequencing by synthesis was performed with pairedend reads of 166 nt and 136 nt using the MiSeq System (Illumina, San Diego, CA, USA).

The data analysis was performed with the “CLC Genomics Workbench” software (QIAGEN Bioinformatics) and the following steps: quality filtering of the reads, merge of paired-end reads, and Motif Search comparing with the known barcodes for each inoculum with a precision limit of 100%. The percentage of every strain was calculated along the experiment. Those strain with a higher colonization load or those than were detected after antibiotic treatment removal were selected for further experimentation. NGS assays design and analyses was performed at the Unidad de Genómica of the UCM.

### 2.5. C. albicans Filamentation Assay

To analyze the ability to filament, 10^5^
*C. albicans* cells were inoculated in one mL of pre-warmed fetal bovine serum and incubated at 37 °C in an orbital shaker. Samples were taken after 3 h and photographed using a Nikon Eclipse TE2000-U microscope at 100× magnification. Images were captured by a Hamamatsu ORCA-ER CCD camera using AquaCosmos 1.3 software. All images were processed identically and mounted using Adobe Photoshop 7.0.

To analyze the colony morphology of *C. albicans* on Spider medium (1% nutrient broth, 1% mannitol, 0.2% K_2_HPO_4_, and 1.5%agar), 50–100 yeast cells were plated on Spider plates medium, incubated at 37 °C for 5 days before pictures were taken.

### 2.6. Supplementary Material and Methods

#### 2.6.1. Susceptibility Assays

Drop tests for susceptibility/resistance assays were performed by spotting 5 µL drops containing cell suspension from 10^5^ to 10^2^
*C. albicans* cells onto solid YPD plates supplemented with different concentrations and compounds. Plates were incubated for 24 h under different O_2_/CO_2_ concentration.

The white/opaque transition was induced by plating 50–100 CFUs on YPD supplemented with 10 µg/mL phloxine B (Sigma-Aldrich, Darmstadt, Germany) and incubated at 24 °C for 96 h before photographs are taken.

#### 2.6.2. Adhesion Assay

Adhesion to plastic was performed in 24-well flat bottom plates for culture cells. Five hundred cells were added to each well in RPMI 1640 medium and allowed to adhere for 30 min. Medium carrying non adhered cells was spread on YPD for CFUs count. Adhered cells were mechanically removed and spread on YPD for CFUs count. Cell adhesion was expressed as percentage of adhesion (adhered cells * 100/(adhered cells + non-adhered cells)).

HT29 cell line were grown on DMEM medium with high glucose (25 mM) (Gibco, Waltham, MA, USA), 10% of heat inactivated fetal bovine serum and penicillin/streptomycin (100 units/mL of penicillin and 100 μg/mL of streptomycin) in a humidified incubator with 5% CO_2_ atmosphere at 37 °C. Culture medium was changed every 2 days until seeding in a 24-well flat bottom plates for culture cell. When HT29 cultures were confluent the adhesion assay was performed. Culture medium was removed and 1 mL of the same medium lacking serum was added per well. Parallelly, *C. albicans* cells grown overnight in YPD at 37 °C were washed twice with PBS, counted in a Neubauer chamber and resuspended in DMEM medium with high glucose plus penicillin/streptomycin. *C. albicans* isolates were mixed with SC5314-RFP to equal amount and 2 × 10^5^
*C. albicans* cells were added per well. For this, the culture medium was replaced by 1 mL of medium containing the mix of *C. albicans* cells to 2 × 10^5^ cell per mL. Plate was incubated in a humidified incubator with 5% CO_2_ atmosphere at 37 °C for 1 h. Then, culture medium was removed and, *C. albicans* cells attached to the HT29 cell line were recovered adding 500 µL of water plus 0.2% Triton twice and mechanically removed. 100 µL from different dilutions were plated on SD medium to distinguish the two types of colonies and to allow CFUs count. Mixed culture adhesion was expressed as Adherence Relative Index (ARI) that was calculated dividing the percentage of adhered cells from every strain by their percentage in the inoculum.

#### 2.6.3. Phospholipase Activity

Phospholipase activity was determinate by growing *C. albicans* strains on egg yolk containing media and measuring the precipitation zone. Two media were used as some differences have been reported: SEA and MEA [34]. SEA (Sabouraud Egg Agar) medium was prepared as follow: 0.65% Sabouraud’s dextrose, 1–0.5 M NaCl, 0.0055% CaCl_2_ and 2% agar were re-suspended in 900 mL H_2_O_2_ and autoclaved. In parallel, one egg was immersed in 70% ethanol for 3 h to disinfect it, then, the yolk was added to 250 mL 0.2% saline solution and homogenized, 100 mL of this solution were added to the 900 mL medium previously prepared and homogenized [35,36]. MEA (Malt Egg Agar) consist in 2% malt extract agar, 2% agar, 2% dextrose, 1–0.5 M NaCl, 0.005% CaCl_2_ and 1% sterile egg yolk (prepared similarly to SEA medium) [34].

Stationary phase growing cells were resuspended at O.D. 1 and 2.5 µL drop was spotted onto MEA or SEA agar plates. Plates were incubated for 120 h at 30 °C, 37 °C in normoxia or 37 °C in hypoxia before plates were scanned and precipitation halos were measured using the J-image. Each isolate was tested in replicates of three in two different days. The phospholipase activity was defined as the ratio of colony diameter to the diameter of the dense white zone of precipitation around phospholipase positive colonies.

### 2.7. C. albicans Susceptibility Testing

Sensititre YeastOne microdilution methods (Thermo Fisher, Waltham, MO, USA) was used as a commercial colorimetric test for the determination of minimum inhibitory concentrations. To perform the test, a 0.5 McFarland suspension of each yeast strain was prepared and 100 µL were inoculated into a culture medium provided by the manufacturer (Sensititre yeast One broth). Plates containing lyophilized antifungals were rehydrated with inoculum suspension using a multichannel pipette (100 µL in each well). The panels had the following serial dilutions: 0.125–8 µg/mL for amphotericin B, 0.12–256 µg/mL for fluconazole, 0.008–8 µg/mL for voriconazole, posaconazole, micafungin and caspofungin, 0.015–8 µg/mL for anidulafungin, 0.016–16 µg/mL for itraconazole and 0.06–64 µg/mL for 5-fluorocytosine. Plates were covered with adhesive and incubated at 35 °C for 24 h. Sensititre YeastOne uses a colorimetric assay for MIC evaluation, where no growth is detected, and the color changes from blue to pink depending on fungal growth. The MIC values were interpreted according to the breakpoints stated in CLSI M27-S4 [37].

## 3. Results

### 3.1. Identification of Clinical Isolates of Candida albicans with Increased Gut Colonization Load

To identify clinical isolates with an increased ability to colonize the gastrointestinal tract, a competitive assay was performed with 114 barcoded *C. albicans* clinical isolates. To barcode the *C. albicans* strains, different clinical strains isolated from different body localizations (see Appendix A) and identified as *C. albicans* by MALDI-TOF (were transformed with a plasmid carrying a unique and distinguishable sequence as well as the *SAT1* gene that confers resistance to nourseothricin. These barcoded strains were randomly grouped into four pools, each pool consisted of 29–30 strains including a SC5314-barcoded control strain per group (Appendix A). Pools were inoculated by gavage in groups of three mice previously treated with a mix of antibiotics in the drinking water. Stool samples were taken at 0, 7 and 21 days, where antibiotic treatment was then removed, and two more samples were taken at 42 and 60 days’ post-inoculation. Samples were processed for CFUs counting and genomic DNA was extracted in pool from solid plates growth to perform NGS analyses.

As seen in Figure 1, fungal colonization load was similar for all groups in the first 20 days, although greater differences could be observed at day 40 following removal of the antibiotic therapy (at day 21 (Figure 1)). The NGS assay was performed by Illumina sequencing and revealed that most of the inoculated strains were detected in a low relative amount or were even undetectable, including the SC5314-barcoded strain used as reference (Appendix A). In total, 8 out of 114 strains were selected either because they showed a high colonization rate compared with their competitors or because they remained in the intestine after removal of antibiotic in the drinking water of mice. The *C. albicans* selected strains and the body localization from they were isolated are listed in Table 2. Although the number of clinical isolates of *C. albicans* was higher than that of strains obtained from healthy individuals (105 versus 9) in the screening one out of eight strains came from a healthy donor indicating that neither the clinical status of the individuals or body location of isolation (vagina, gut, etc.) was overrepresented in this subset of strains selected.

### 3.2. Selected C. albicans Strains Displayed Enhanced Fitness Compared to SC5314

To confirm that the selected *C. albicans* strains display an enhanced ability to colonize the murine gastrointestinal tract, a competitive assay was performed between each selected strain and an RFP-labelled SC5314 strain [14]. All the selected strains (Table 2) were able to colonize the murine intestine of C57BL/6 mice to higher or similar levels than the control strain SC5314-RFP; moreover, all these strains remained in the intestine for a prolonged period (greater than 20 days) after removal of antibiotic therapy that took place at 17 day (Figure 2 and Appendix A). These results validate our screening while, additionally, allow the identification of *C. albicans* strains that could persist as a commensal without antibiotic treatment.

Selected strains were phenotyped to identify traits that could be responsible for the fitness increase over SC5314. The susceptibility to different types of stress, the ability to adhere to polystyrene or to the human epithelial intestinal HT-29 cell line, the tolerance to bile salts or their phospholipase activity, among other phenotypes, were tested. Significant differences were observed in the susceptibility to bile salts and their phospholipase activity in vitro (Table 3 and Appendix A). While no differences were detected when osmotic stress (NaCl), inhibitor of the electron transport chain or adhesion to epithelial intestinal HT-29 cell line were tested, two out of eight strains displayed an enhanced tolerance to menadione (Pract.2018-1 and CaHG14) and one strain (CaORAL3) displayed decreased adherence to polystyrene. No differences were observed among strains when other oxidative agents were analyzed. All the analyzed strains were heterozygous for the mating type and did not stain with phloxine B. We conclude that the enhanced ability to colonize the murine gut does not correlate with any of the phenotypes tested and may, therefore, rely on other characteristic(s) not yet determined in our studies or in a combination of them. The summary of the most relevant phenotypic features is shown in Table 3.

The Minimum Inhibitory Concentration to different antifungals was tested following the CLSI protocols (Appendix A). All the selected *C. albicans* isolates were sensitive to the drugs tested according to CLSI standards.

### 3.3. Strains with Increased Fitness Are Able to Form True Hyphae

The adaptation of *C. albicans* to the murine gastrointestinal tract has been correlated with the loss of the ability to filament via mutations in the *FLO8* gene [25]. To determine if the enhanced fitness of the selected *C. albicans* strain correlated with alterations in the yeast-to-hypha transition, we tested the ability to filament in either in liquid or solid media. Stationary phase cultures in YPD at 37 °C were inoculated in serum to 10^5^ cells per mL and incubated at 37 °C for 3 h (Figure 3A). All the strains were able to form true hyphae in serum; filaments could also be observed in the pre inoculum (YPD liquid medium) indicating that these strains retained the ability to filament.

In addition, fifty to one hundred CFUs were spread on Spider medium and incubated at 37 °C for 5 days to observed agar invasion (Figure 3B). Six out of eight strains were able to invade on Spider medium plates. Consequently, the enhanced ability to colonize murine gut does not correlate with the deficiency to invade or with significant defects in morphogenesis.

### 3.4. CaORAL3 Strain Still Requires Antibiotic Treatment to Colonize Murine Gut

The CaORAL3 strain was selected for further characterization and the ability to colonize naïve mice was analyzed. CaORAL3 was unable to colonize the mouse gut without previous antibiotic therapy. As shown in Figure 4A, when 10^7^
*C. albicans* cells were inoculated intra-gastrically in non-antibiotic treated mice, low CFUs were recovered from stools at days 1 and 7 post-inoculation but not at subsequent days. A similar dose of inoculum enabled colonization in mice treated with antibiotics for 4 days before *C. albicans* gavage, the antibiotics treatment was maintained for 9 days’ post-inoculation (Figure 4B). The removal of antibiotics did not alter the colonization level that remained invariable for as long as 44 days. To check whether these high levels are needed for the stable maintenance in the antibiotic-regime, we forced the reduction of fungal cells by oral administration of nystatin and fluconazole for 4 days; this resulted in a severe reduction of fungal loads; however, the remaining yeast cells were able to eventually recover previous fungal levels after stop of antifungal treatment (Figure 4C).

These results indicate that *C. albicans* colonization occurs in the new gastrointestinal conditions (that is after antibiotic treatment) but do not provide information about the actual role of fungal cell in promoting these conditions. Therefore, a set of experiments were designed where CaORAL3 was inoculated 3 days before (B), just after (J) or 7 days after (A) a 10-days antibiotic treatment. In other words, three groups of mice were treated with the same antibiotic regimen for 10 days (days 1–10) and CaORAL3 cells were introduced by gavage at day 7 (B), day 10 (J) or day 17 (A). The three groups exhibited comparable colonization loads, although a slightly higher colonization load was observed in B group indicating a partial-albeit not essential- contribution of the fungi for inducing the permissive conditions (Figure 5B). These experiments suggest that antibiotics cause microbiota alterations that allow the colonization by CaORAL3 and that this alteration remains stable for at least 7 days after removal of the treatment.

### 3.5. Changes in the Intestinal Microbiota Allow CaORAL3 Colonization

To analyze the effect of naïve microbiota on *C. albicans* CaORAL3 colonization, we performed two different approaches: (1) gavage with extracts of feces and (2) co-housing with naïve mice. In the first approach we compared two identical groups of mice colonized with CaORAL3 where the antibiotics were retired at least 30 days before the experiment; both groups presented similar fungal loads (≈10^6^ CFU/g). The first was inoculated intra-gastrically twice (separated 48 h) with feces from naïve mice (group F) while the second one was maintained as control (group C) (Figure 6A). Naïve microbiota led to a ≈100-fold decrease in CaORAL3 colonization load; however, this strain was able to persist in the gut at above threshold levels for more than 30 days after naïve microbiota inoculation. The second approach used co-housing, therefore allowing a “natural” interchange of microbiota (including fungi) among mice due to coprophagia [14]. Two naïve mice (group N) were introduced in a cage with a group of 2 mice colonized with CaORAL3, where the antibiotics were retired at least 30 days before (group H) (Figure 6B). Acquisition of conventional microbiota resulted in a slow decreased of fungal levels in group H up to a 10-fold reduction, that remained stable for more than 3 weeks. Surprisingly, CaORAL3 strain was detected also in naïve mice (group N). From the first day of co-housing group N showed a stable *C. albicans* colonization, although the levels in the new-introduced mice were lower than in the previously inoculated mice (around 10^5^ CFU/g) (Figure 6B). However, CaORAL3 strain samples obtained from group H feces were not able to colonize naïve mice after in vitro growth and inoculated by gavage. These results suggest that intestinal microbiota is important to control *C. albicans* colonization but once *C. albicans* has been established as a commensal, a fecal transplant is not enough to remove CaORAL3 strain from the intestine.

### 3.6. CaORAL3 Is Able to Colonize Murine Gut Previously Colonized with C. albicans

We next determined whether CaORAL3 could impede the colonization of a wild type reference strain. To analyze this, we introduced intra-gastrically an inoculum of 10^7^ RFP-labeled SC5314 yeast cells from an in vitro culture in mice previously colonized by CaORAL3 and where antibiotics were retired 39 days before (Figure 7). As observed in Figure 7A, the SC5314-RFP strain was unable to establish a long-term colonization, although medium fungal loads (10^4–6^) were achieved during the first 10 days. In a parallel control group, CaORAL3-RFP obtained from an in vitro culture was inoculated in mice previously colonized by CaORAL3; the RFP labeled strain also showed lower loads at the beginning, but it ended up reaching roughly the same high colonization loads than already colonizing CaORAL3 (Figure 7B). *Post mortem* analyses revealed that CaORAL3 and CaORAL3-RFP displayed the same distribution along the gastrointestinal tract, indicating that the strain inoculated in second place was able to compete and occupy the same niche as the previously established strain (Figure 7C). In a separate experiment, we introduced cells of CaORAL3-RFP strain in mice previously colonized by CaHG43 strain where the antibiotics were retired 39 days before (Figure 7D). Again, CaORAL3 was established in the gut indicating that this strain can compete even when other *C. albicans* strains are already occupying the intestine.

## 4. Discussion

*C. albicans* remains as the most prevalent fungal pathogen in humans [1] probably because its derived infections are mainly endogenous. The intestine (mainly the cecum) contains elevated number of *C. albicans* cells that can cause disseminated candidiasis when host defenses are compromised. Experimental animals are essential models to analyze the virulence of *C. albicans* as well as the determinants of adaptation to the commensal state [9]. However, laboratory mice do not carry *C. albicans* as a natural commensal in their intestine. This fact requires us to manipulate the natural gut microbiota to enable colonization via the use of gnotobiotic mice or mice where the endogenous bacterial components of the microbiota have been significantly reduced by antibiotic administration. Both models lead to similar colonization loads (estimated by CFUs counting) in stools or intestinal content [11] although differences were observed in the morphology and localization of the fungal cells. An advantage of the second model is that it can be conveniently manipulated (interrupted or altered) and mimics a frequent risk factor for human candidiasis [14,23,27]. Using this model, we have screened a collection of clinical isolates by an NGS-barcoding strategy analyzing up to 29–30 different strains in a single group of mice and selecting 2–3 strains per group with high ability to colonize the mouse gut.

The development of animal models to study the commensalism to molecular level revealed the relevance of different genes and metabolic pathways in this process. Overexpression of the transcription factor Crz2 leads to a higher tolerance to bile salts and acid pH under hypoxia [38], features that could explain the improved ability to colonize murine gut. Similarly, the lack of the MAP kinase Hog1 leads to bile salts and oxidative stress susceptibility which correlates with the inability to establish as a commensal in the murine gut [14]. Following the idea that certain phenotypic characteristics could predict the greater or lesser capacity to colonize the gut, the collection of clinical isolates was analyzed before and after the commensalism assay. Susceptibility to bile salts, oxidative stress, electron transport chain inhibitors or phospholipase activity, among other phenotypes, were checked. Differences were observed in the susceptibility to different stresses such as bile salts or menadione, which revealed the variability between the different isolates of *C. albicans* distributed in the population and, possibly the adaptation to specific niche or microbiota. None of the strains were homozygous for the mating type nor displayed GUT or opaque morphologies [22,27]. Although many of these characteristics have been previously related to alterations in the ability to colonize murine gut [14,26,27], in our phenotyping not a single trait was identified as predictor of the higher fitness in commensalism. This may indicate that commensalism is a multifactorial process lying on different attributes. Moreover, we focus on just a few of many chemicals and stimuli that are part of the intestinal environment. In any case, transcription factors or signaling elements seem to be the key for the establishment of *C. albicans* as a commensal.

The adaptation of *C. albicans* to murine gut by long-term gastrointestinal colonization in germ-free antibiotic treated mice allowed the selection of non-virulent non-filamentous *C. albicans* cells [25]. This suggests that dimorphic transition is not crucial for *C. albicans* to colonize murine gut in agreement with results obtained by different approaches [11,23,39]. The clinical strains analyzed in the present work retained the ability to filament whatever the origin from which they were isolated. This fact suggests that evolution in murine or human guts force to develop different phenotypes. The microbiota seems to play an essential role in this process. It is also evident the differences between mice and humans and the necessity to develop animal models closer to human in order to be able to extrapolate the observations made in the mouse.

Interestingly, the CaORAL3 strain keeps colonizing the gastrointestinal tract after antibiotics are removed; therefore, in a way quite more similar to the human case. This strain is able to establish as a commensal with just a short-term antibiotic treatment. Even more, it can establish in the murine gut of non-antibiotic treated mice co-habiting with mice previously colonized or colonize mice previously colonized by other *C. albicans* strain. Previous studies indicate that *C. albicans* requires a time (15–21 days) to adapt to murine gut, these adapted cells displayed a higher fitness and were able to impair the establishment of non-adapted *C. albicans* cells [40]. This study was performed with derived strains from the CAF2 wild type strain and suggests that during prolonged stays in the murine gut transcriptomic and/or epigenetic changes must be taken place in order to improve adaptation to this niche. CaORAL3 strain behaved as an adapted strain. Remarkably, this strain remains (although at lower levels) in the murine gut even when mice were inoculated (natural or artificially) with non-antibiotic treated murine microbiota. All these characteristics make this strain an important starting point for new experimental models in which the role of probiotic or human microbiota and its interaction with *C. albicans* can be analyzed in the murine gut.

Antibiotic treatment is required to colonize the murine gut even for CaORAL3 strain. In our assays the changes in the microbiota that allowed *C. albicans* establishment remain for at least 7 days after antibiotic treatment removal as indicate the fact that CaORAL3 was able to colonize gastrointestinal tract in mice without antibiotic administration for 7 days before *C. albicans* gavage. Antibiotic treatment is usually correlated with an important decrease of the microbiota in amount and in diversity [41]. However, our results show that the *Candida* permissive microbiota can be somehow transferred to cohoused untreated mice, therefore suggesting the presence of certain refractory and transmissible microorganisms that may promote those permissive conditions. These microorganisms may also impair the effective acquisition of regular microbiota, since inoculation with it causes a reduction on the *C. albicans* load level but does not remove it from the gut. Previous publications where the role of the microbiota in *C. albicans* colonization was analyzed, reported that commensal bacteria act as a barrier being critical for *C. albicans* colonization [41]. To our knowledge, all the studies previously reported required the maintenance of the antibiotic treatment, otherwise *C. albicans* became removed from the murine gut. Among the great diversity of the bacterial microbiota *Lactobacillus rhamnosus* [42], *Blautia producta* and *Bacteroides thetaiotamicron* [43] were identified as critical to prevent *C. albicans* colonization, but none as relevant for favor fugal establishment.

Deeper characterization of the microbiota is required to identify those filum, genus or species that allow CaORAL3 colonization. Anyway, the identification of 8 out of 114 *C. albicans* strains able to remain in the gut after stopping the antibiotic treatment suggests that CaORAL3 is not the only strain with improved fitness in gut and insinuates the existence of an important diversity among *C. albicans* commensal strains. The sequencing of the complete genomes of these strains and their comparative analysis may give us an idea of the genetics reorganizations or mutations involved in the increased fitness in the mouse gastrointestinal tract. Thus, microbiota and the immune system are relevant, but *C. albicans* diversity must be also taken in consideration when addressing these types of studies.

*C. albicans* has been traditionally judged as a human pathogen; then, the virulence and the analysis of the virulence factors has been the focus of attention for years. Nevertheless, the fact that *C. albicans* is the main permanent fungus in the human microbiota unfolds new perspectives and open the question of its role in the microbiota. Wild type *C. albicans* strains able to colonize murine gut by minimally altering the microbiota may provide a more physiological model to analyze the role of the commensal bacteria, host immune system and *C. albicans* interactions.

## Figures and Tables

**Figure 1 jof-07-00695-f001:**
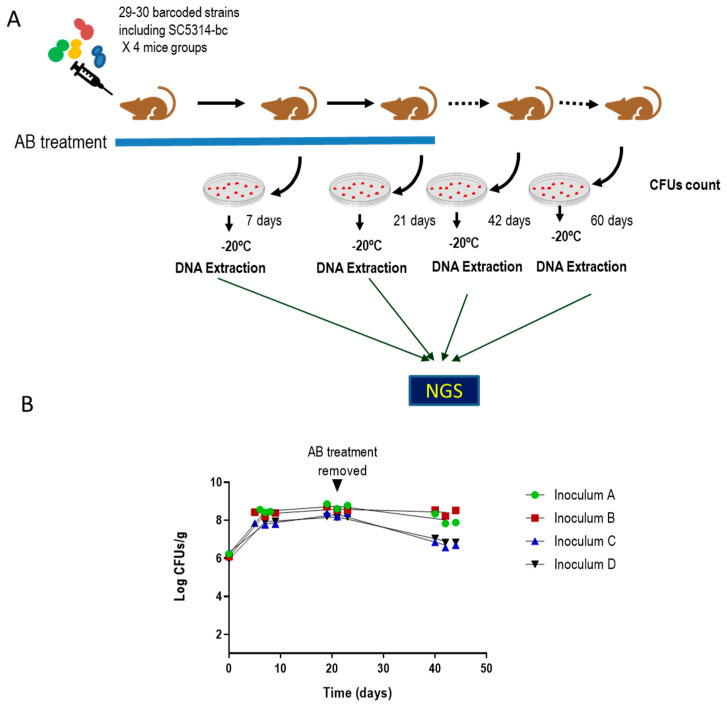
Assay to identify *C.*
*albicans* clinical isolates with increased ability to colonize the murine gut. (**A**) Schematic representation of the assay. (**B**) Fungal colonization load of *C. albicans* strains in pool. Inoculum A, B, C and D correspond to the four pools of *C. albicans* barcoded strains (Appendix A). Each *C. albicans* strains pool was inoculated intragastrically in 3 mice with a partially depleted microbiota. Antibiotic treatment was maintained for 21 days after *C. albicans* gavage then, it was removed. Stools were taken at different time points and processed for CFUs counting. Data are represented and expressed as log CFUs per stool gram versus time (days). Each replicate is shown and each line represents the mean of the data.

**Figure 2 jof-07-00695-f002:**
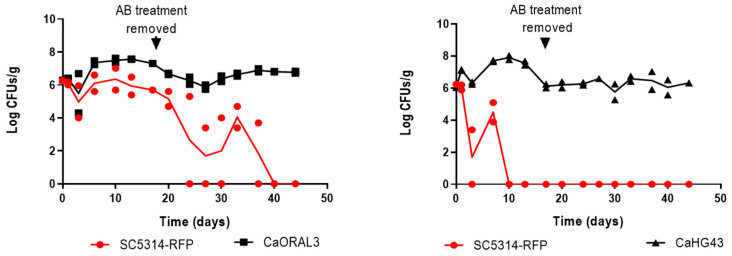
Competitive colonization assays. CaORAL3 and CaHG43 strains were inoculated in a 1:1 ratio with SC5314-RFP in 3 mice by gavage. The time course of fungal colonization load was followed by CFUs counting of stools in SD-chloramphenicol medium. Antibiotic treatment was removed at day 17. Appendix A shows the colonization load for the other six selected *C. albicans* strains in competition with the SC5314-RFP.

**Figure 3 jof-07-00695-f003:**
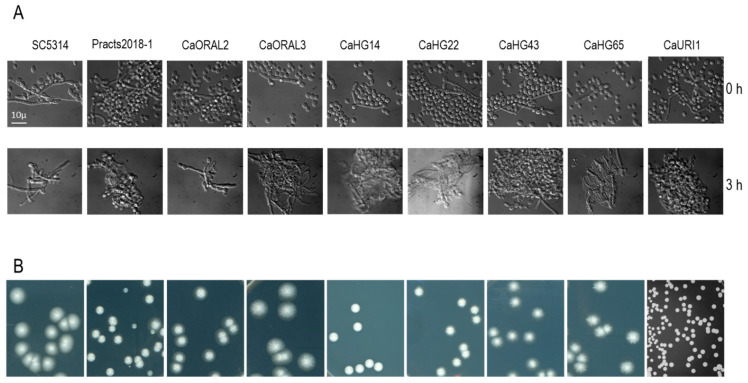
Ability to perform the yeast-to hypha transition in liquid and on solid media. (**A**) The strains indicated in the upper row were grown in YPD overnight at 37 °C (time 0) and refreshed in pre-warmed serum. Samples were taken 3 h later and photographed under the microscope. (**B**) In total 50 to 100 cells from cultures in the stationary phase were spread on Spider medium and incubated for 5 days at 37 °C.

**Figure 4 jof-07-00695-f004:**
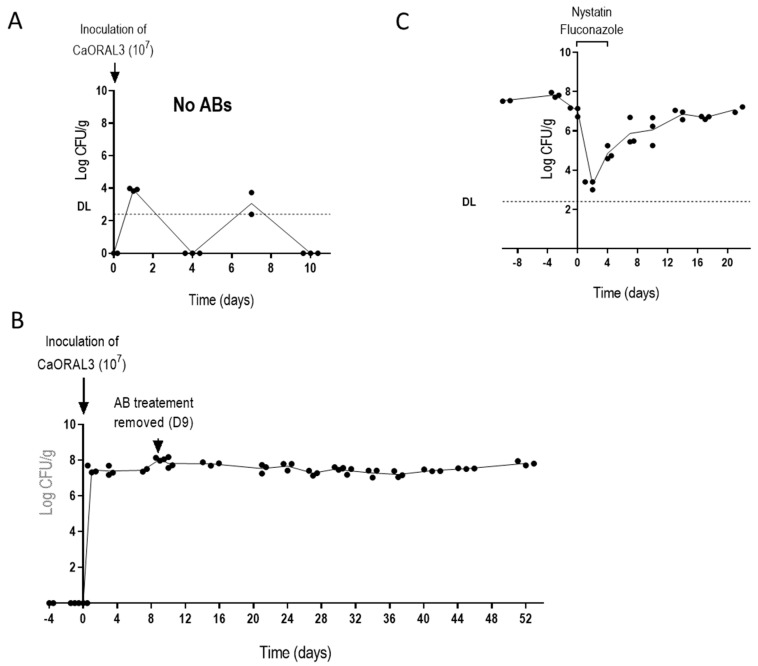
Effect of the antibiotic treatment on CaORAL3 colonization. (**A**) A suspension of CaORAL3 cells was inoculated in non-antibiotic treated mice. (**B**) CaORAL3 cells were inoculated in Ab-treated mice 4 days prior plus 9 days after *C. albicans* inoculation. Colonization rate was followed in time and expressed as Log CFUs per g of stools. (**C**) Effect of nystatin and fluconazole administration over CaORAL3 colonization rate.

**Figure 5 jof-07-00695-f005:**
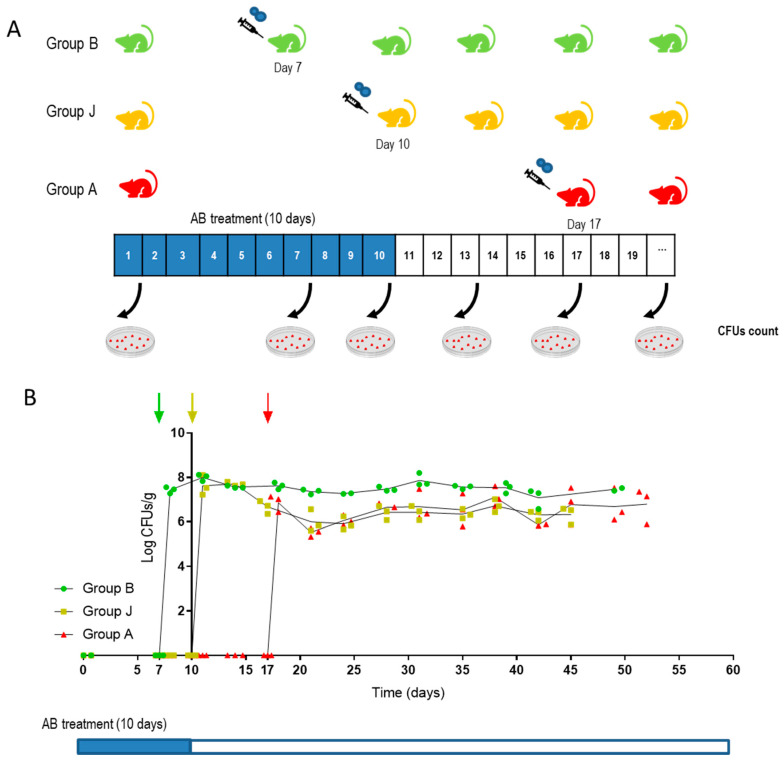
Determination of the minimum antibiotic regimen to allow gut colonization (**A**) Schematic representation of the assay, three groups of mice were treated with antibiotics for 10 days, 10^7^ CaORAL3 cells were inoculated 3 days before (**B**), just after (J) or 7 days after (**A**) the 10-days antibiotic treatment. Stool samples were taken at different time and processed for CFUs count. Data are shown as Log CFU/g of feces (**B**). Arrows indicate *C. albicans* inoculation for each group.

**Figure 6 jof-07-00695-f006:**
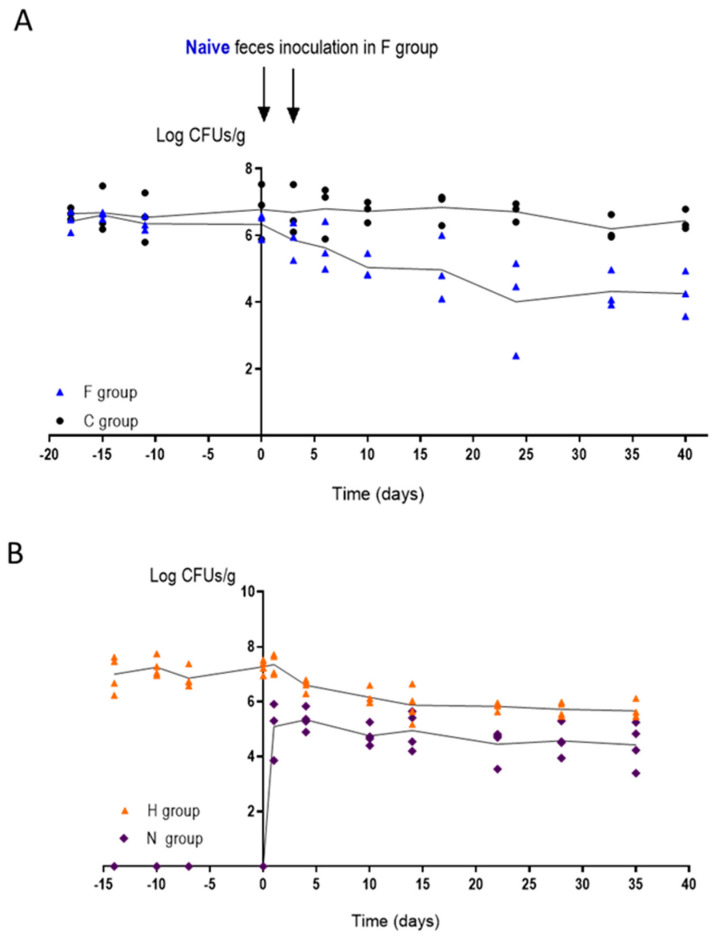
Influence of the microbiota in mice stably colonized with CaORAL3 strain and without antibiotic treatment for 30 days. (**A**) Feces from naïve mice was inoculated in F group and colonization load was followed in time. Group C was followed as control. (**B**) Naïve mice (N group) were introduced in the cage of mice previously colonized by CaORAL3 (H group) and colonization load was followed in time. Results of two independent assays are shown in the graph.

**Figure 7 jof-07-00695-f007:**
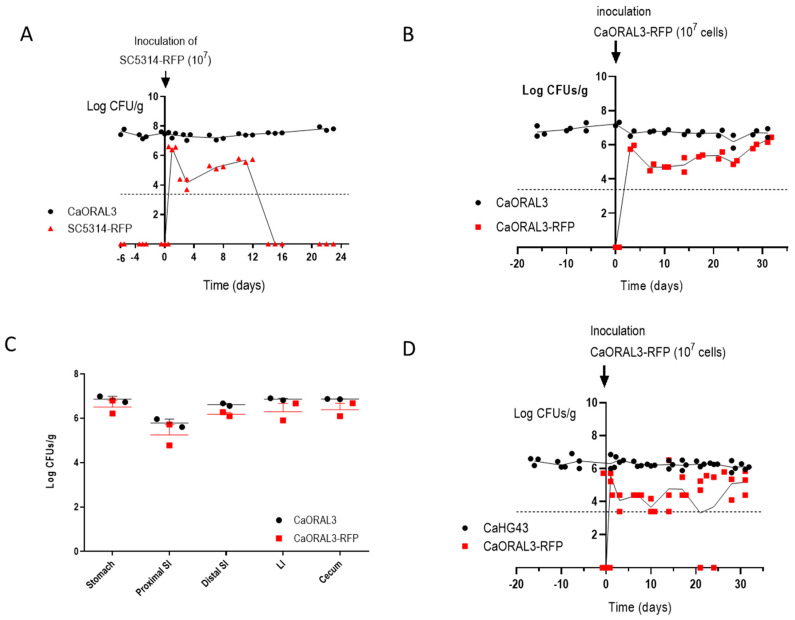
Competition between CaORAL3 and other *C. albicans* strains in the murine commensalism model. (**A**) 10^7^ SC5314-RFP cells were inoculated in mice previously colonized by CaORAL3 cells and without antibiotic treatment for 39 days. Colonization load was followed in time by spreading stool suspension on SD-chloramphenicol medium to distinguish white/red colonies. (**B**,**D**) 10^7^ CaORAL3 cells tagged with RFP were inoculated in mice previously colonized by (**B**) CaORAL3 or (**D**) CaHG43 strains and without antibiotic treatment for 39 days. Colonization load was followed similarly. (**C**) At the end of the experiments, mice from groups in (**B**) were euthanized and the amount of each strain was quantified in the different portions of the gastrointestinal tract.

**Table 1 jof-07-00695-t001:** *C. albicans* strains used in this work.

Strain Name	Strain Background and Genotype	Reference
SC5314-RFP	SC5314 *ADH1*/*adh1*::*SAT1-dTOM2*	This work
CaORAL3-RFP	CaORAL3 *ADH1*/*adh1::SAT1-dTOM2*	This work

**Table 2 jof-07-00695-t002:** *C. albicans* strains selected in our competitive pool commensalism assay.

Strain	Origin	Sample
Practs 2018-1	Commensal	Stool culture
CaORAL2	Hospital	Oral cavity
CaORAL3	Hospital	Oral cavity
CaHG14	Hospital	Bronchoalveolar lavage
CaHG22	Hospital	Pharyngeal exudate
CaHG43	Hospital	Pharyngeal exudate
CaHG65	Hospital	Urine culture
CaURI1	Hospital	Urine culture

**Table 3 jof-07-00695-t003:** Summary of phenotypic characteristic displayed by the selected *C. albicans* isolated in comparison to standard SC5314-RFP. Only differences are shown in the table. S sensible, R resistant, <adhesion to polystyrene compared to SC5314-RFP, ↓ low phospholipase activity, +++ very high phospholipase activity, ++ high phospholipase activity, ns means no significant differences were observed.

Strain	Invasiveness in Spider Medium	Adhesion to Polyestyrene	Bile Salts Sensitivity	Menadione	Phospholipase Activity (SEA)
Pract. 2018-1	n.s.	n.s.	R	R	n.s.
CaORAL2	n.s.	n.s.	R	n.s.	n.s.
CaORAL3	n.s.	<	R	n.s.	n.s.
CaHG14	Non invasive	n.s.	n.s.	R	n.s.
CaHG22	n.s.	n.s.	S	n.s.	↓
CaHG43	n.s.	n.s.	n.s.	n.s.	+++
CaHG65	n.s.	n.s.	n.s.	n.s.	↓
CaURI1	Non invasive	n.s.	n.s.	R	++

## Data Availability

The data presented in this study are available in Appendix A.

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
