# Peer review of "Identification of Clinical Isolates of Candida albicans with Increased Fitness in Colonization of the Murine Gut"

_jof, 2021, doi:10.3390/jof7090695_

Round 1
Reviewer 1 Report
Alonso et al have gathered isolates from patients with a variety of candidiasis diseases. They have shown that one isolate (CaORAL3) persisted more in lab experiments with mice. This strain seemed to also not require anti-microbials for colonization. Phenotype studies of strains using standard tests for virulence in C. albicans did not reveal any advantages for this strain. So, no mechanism is stated. A future for this study, the CaORAL3 strain may be useful in a correlation of antibacterial usages in patients.
Thoughts:
All strains tested caused human disease. That seems to be a bottom-line correlate. Does this make the current data less significant?
The discussion is fairly long and should be rewritten clearly to pose ideas about the differences in CaORAL3.
It seems to me that if phenotypes were to be used to identify isolate differences, then there is much more of this to do. I do not recommend that study, too many virulence factors. An antibiotic could have eliminated an especially inhibitory bacterium to CaORAL3, allowing its persistence. Perhaps bacterial populations could be studied in the persisting isolate versus other strains that did not persist. Thus, of the mixture of isolates studied, only one persisted that could be a reflection of what antibiotic did to what bacteria in the gut.
In several experiments, the control 5314 strain looks different in persistence, sometimes no and sometimes yes.
Table 4 data should be briefly stated only in the text. If there are differences, that is reflected in a greater sensitivity of CaORAL3. There is no take-home thought with these experiments.
Fig 3A. CaORAL3 colonies are larger, any reason for that?
Fig 4B, no control strain?
Author Response
Reviewer comment: All strains tested caused human disease. That seems to be a bottom-line correlate. Does this make the current data less significant?
Authors answer: Most of the strains used in the initial screening were isolates from fungal infection collected from hospitals in blood samples; nevertheless, commensal strains isolated from feces were also tested. Phenotypic analysis revealed no evident differences between clinical isolates or commensal C. albicans strains, suggesting in principle that there are no major differences between both types of strains. This, in fact, is one of the bottom lines of this work as raised by the referee.
Reviewer comment: The discussion is fairly long and should be rewritten clearly to pose ideas about the differences in CaORAL3.
Authors answer: Following the reviewer suggestion, we have shortened the discussion. We removed part of the discussion about the relevance of bile salts in the microbiota composition and kept the description of CaORAL3 behavior. We are open to new changes if the reviewer considers it appropriate.
Reviewer comment: It seems to me that if phenotypes were to be used to identify isolate differences, then there is much more of this to do. I do not recommend that study, too many virulence factors.
Authors answer: We agree with the reviewer; there are many possible virulence factors so trying to find the proper one(s) may be difficult. In any case, we selected those virulence traits that could, in principle, be related to pathogenesis, such as bile salt, production of phospholipase or adhesion trying to avoid others that could be hardly related to the mechanisms of pathogenesis.
Reviewer comment: An antibiotic could have eliminated an especially inhibitory bacterium to CaORAL3, allowing its persistence. Perhaps bacterial populations could be studied in the persisting isolate versus other strains that did not persist. Thus, of the mixture of isolates studied, only one persisted that could be a reflection of what antibiotic did to what bacteria in the gut.
Authors answer: The analysis of gut microbiota is an interesting point for further analysis. Moreover, we think that this microbiota is modified not only by the antibiotic treatment but by the C. albicans strain itself. This hypothesis is introduced in the discussion, but NGS analysis of microbiome is a large demanding work that was out of the scope of this work.
Reviewer comment: In several experiments, the control 5314 strain looks different in persistence, sometimes no and sometimes yes.
Authors answer: The reviewer is right; the persistence of SC5314 control strain depends on the strain used in the competition commensalism assay. This has been highlighted in the manuscript. The following sentence has been included: “It is remarkable that the behavior of the control strain, SC5314-RFP depended on the strain to which is compared in the competition commensalism assay, suggesting differences in the fitness for these strains”. We think, as well that this type of assays truly reflects what should be a fitness competition.
Reviewer comment: Table 4 data should be briefly stated only in the text. If there are differences, that is reflected in a greater sensitivity of CaORAL3. There is no take-home thought with these experiments.
Authors answer: Following the reviewer suggestion, table 4 has been removed from the manuscript and included as supplementary information.
Reviewer comment: Fig 3A. CaORAL3 colonies are larger, any reason for that?
Authors answer: Figure 3B shows the colonies on Spider medium after 5 days of incubation at 37ºC, CaORAL3 and the control strain SC5314, displayed very similar colonies in size and an evident invasive growth beyond the colony borders.
Reviewer comment: Fig 4B, no control strain?
Authors answer: Commensalism assays can be performed in competition (for example figure 1 and 2) or with single strains. In figure 4, assays were performed with CaORAL3 in solitary. These assays were performed to study the behavior of this strains in the commensalism model. No control strain is required.
Reviewer 2 Report
JoF-1337912
A very interesting paper; however, experiments presented seem to be over-sophisticated and not correctly designed therefore messages and conclusions are not fully supported by data obtained.
The animal model chosen to study Candida colonization after a very wide use of the antibiotics is highly artificial and far from natural processes. It would be of benefit to such a study to use germ-free mice neonates instead of adult mice with antibiotic-modified gut microbiota. Moreover, the authors, although used NGS technology for strain identification did not applied this powerful tool to check gut microbiota in studied mice along the experiment(s) to follow its alterations after antibiotic treatment. Such an approach should also help to explain visible discrepancies among strains and animals.
Line 41/42 – This sentence contains phrases not used to describe ecological mechanisms in gut microbiota like as “main origin” or “internal reservoir”.
Line 402-403 – All mice kept in the same cage (co-haused) have identical microbiota since rodents are feces eaters (as the authors already mentioned); and therefore, including group 1) was not necessary. The results reflected this mechanism of the strain transmission.
Para 3.7 – the protocol of this experiment is definitely too complicated and therefore its results are very difficult to understand and to explain.
Line 391-392 – colonization by C.albicans for 7 days seems to be rather short.
Line 563 – “Thus, microbiota and immune system are relevant but C. albicans diversity must be also taken in consideration when addressing these types of studies”. It is a platitude.
Line 569 – 571 – “Wild type C. albicans strains able to colonize murine gut by minimally altering the microbiota may provide a more physiological model to analyze the role of the commensal bacteria, host immune system and C. albicans interactions”. – It is simply not true: this model is far from physiology of the gut. Moreover, the authors several times write about host immune system, although they have not studied it.
Author Response
Reviewer comment : The animal model chosen to study Candida colonization after a very wide use of the antibiotics is highly artificial and far from natural processes. It would be of benefit to such a study to use germ-free mice neonates instead of adult mice with antibiotic-modified gut microbiota. Moreover, the authors, although used NGS technology for strain identification did not applied this powerful tool to check gut microbiota in studied mice along the experiment(s) to follow its alterations after antibiotic treatment. Such an approach should also help to explain visible discrepancies among strains and animals.
Authors answer: As it is mentioned in the introduction, C. albicans is not a common member of the laboratory mice microbiota; as a consequence, antibiotic treatment, the use of germ-free mice or neonates are required for colonization. While we agree that the model is artificial, we believe the use of germ-free mice is much more artificial as such condition is not found in humans. Furthermore, broad antibiotic treatment therapy may result in the overgrowth of C. albicans in the human intestine and therefore, be considered a risk factor for fungemias (Zaborin A et al,. mBio. 2014; 5(5):e01361–14. and Alonso-Monge R et al, PLoS Pathog. 2021;17(7):e1009710). Antibiotic treated mice are, at least, a model to be considered for these studies. This is, in fact, a chosen model for many of these studies. The reviewer is correct that NGS technology is very powerful and has been only used here to identify C. albicans. The study of the microbiota (by NGS techniques) is an interesting question to be analyzed in the future but also a complex process that would require a different experimental set up and was out of the scope of this work. We focused, instead, in finding whether the origin of isolation of a particular strain had a relationship with its fitness attributes in this model.
Reviewer comment: Line 41/42 – This sentence contains phrases not used to describe ecological mechanisms in gut microbiota like as “main origin” or “internal reservoir”.
Authors answer: The reviewer is right. Then, we have rephrased the sentence as follows: “The main cause of candidiasis are commensal strains being the gut an important source of infection”
Reviewer comment: Line 402-403 – All mice kept in the same cage (co-haused) have identical microbiota since rodents are feces eaters (as the authors already mentioned); and therefore, including group 1) was not necessary. The results reflected this mechanism of the strain transmission.
Authors answer: The reviewer is right; CaORAL3 strain was able to colonize naïve mice from colonized mice by co-habiting. We maintain a separate cage with colonized mice as control since this type of experiments require long period of time without antibiotics and the behavior of CaORAL3 in such experimental conditions was uncertain.
Reviewer comment: Para 3.7 – the protocol of this experiment is definitely too complicated and therefore its results are very difficult to understand and to explain.
Authors answer: We agree that fecal transplant analysis are not easy to design and its interpretation is complex. These experiments were designed to analyze the role of naïve murine microbiota in the colonization of CaORAL3 strain since antibiotic treatment modifies the murine gut microbiota and this alteration allows C. albicans colonization. Therefore, we inoculated naïve microbiota in mice where CaORAL3 was already established. As naïve gut microbiota was not able to remove CaORAL3 from the murine gut, this suggests that either fungal or host derived factors (beyond microbiota) also influence CaORAL3 colonization.
Reviewer comment: Line 391-392 – colonization by C.albicans for 7 days seems to be rather short.
Authors answer: Feces were analysed for longer times but CaORAL3 strain was only detected at day 1 and 7.
Reviewer comment: Line 563 – “Thus, microbiota and immune system are relevant but C. albicans diversity must be also taken in consideration when addressing these types of studies”. It is a platitude.
Authors answer: The reviewer is right, but we wanted to remark this idea since experiments try to simplify a really complex system in which all the elements are relevant.
Reviewer comment: Line 569 – 571 – “Wild type C. albicans strains able to colonize murine gut by minimally altering the microbiota may provide a more physiological model to analyze the role of the commensal bacteria, host immune system and C. albicans interactions”. – It is simply not true: this model is far from physiology of the gut. Moreover, the authors several times write about host immune system, although they have not studied it.
Authors answer: This sentence was written as a general conclusion or a desirable future commensalism model. We did not pretend to say that our model is the perfect one. If reviewer considers that the sentence should be removed we have no problem doing so.
We have discussed before our model and indicated that there is no perfect model. For example, we do not believe in germ-free mice as a valid model, as this is far away from reality; however, it can provide interesting data with its limitations, as all models do.
Again, antibiotic treatment is artificial in mice but it can be clinically relevant in humans. The sentence tries to indicate that if we obtain strains that colonize the mice gut without altering the microbiota, this opens a wide array of possibilities: understanding host factors or fungal virulence factors or even treatments that counterbalance C. albicans colonization via several methodologies (e.g. genomics). One important aspect of the antibiotic-model mice is that fungal loads in the gut are very high, and this is not probably very physiological in terms of human colonization. However, we have used a fecal transplant method (feces from naïve mice or co-housing) to make the model more clinically relevant. We think that the strain isolated and characterized in the present work will allow the establishment of a gut colonization murine model closer to what occurs in human microbiota. The murine immune system is another factor to consider and study and clearly different from the human, but addressing this is out of the scope of the work although will be done in the future.
Round 2
Reviewer 1 Report
No additional changes required.
Author Response
Thank you very much for your work.
Reviewer 2 Report
The para (line 332-334 in red) should be rejected since it is rather difficult to understand and addressed to the control strain thus brings nothing important.
Author Response
Response to reviewer 2
Point: The para (line 332-334 in red) should be rejected since it is rather difficult to understand and addressed to the control strain thus brings nothing important.
Response: I am very sorry but the numbering of lines must vary between the different versions of the manuscript (docx or pdf). I don't know what paragraph you are refering. Could you please reproduce the exact paragraph in order to delete it in the final version of the manuscript? I will really apprecite it. I apologize for the inconvenience